



# The contribution of residential wood combustion to the PM$_{2.5}$ concentrations in the Helsinki Metropolitan Area

Leena Kangas[1], Jaakko Kukkonen[1,2], Mari Kauhaniemi[1], Kari Riikonen[1], Mikhail Sofiev[1], Anu Kousa[3], Jarkko V. Niemi[3], Ari Karppinen[1]

[1]Finnish Meteorological Institute, Erik Palménin aukio 1, P.O.Box 503, FI-00101 Helsinki
[2]Centre for Atmospheric and Climate Physics Research, and Centre for Climate Change Research, University of Hertfordshire; College Lane, Hatfield, AL10 9AB, UK
[3]Helsinki Region Environmental Services Authority, Ilmalantori 1, FI-00240 Helsinki

*Correspondence to*: Leena Kangas (leena.kangas@fmi.fi)

**Abstract.** This article has investigated the contribution of residential wood combustion (RWC) to the fine particulate matter (PM$_{2.5}$) concentrations in the Helsinki Metropolitan Area (HMA) for six years, from 2009 to 2014. We have used the PM$_{2.5}$ concentrations measured at eight air quality measurement stations. The dispersion of emissions on an urban scale was evaluated with multiple source Gaussian dispersion models UDM-FMI and CAR-FMI, and on a regional scale using the chemical transport model SILAM. The overall agreement of the predicted concentrations with measurements of PM$_{2.5}$ was

good or fairly good for all stations and years, e.g., at the permanent residential station the daily average values of the index of agreement ranged from 0.69 to 0.81, and the fractional bias values ranged from −0.08 to 0.11, for the considered six years. Both the measured and predicted daily averaged concentrations showed increasing trends towards the lower temperature values. The highest predicted annual averaged concentrations in the region occurred in the vicinity of major roads and streets, and the suburban residential areas, to the northwest, north and northeast of the city centre. The average

concentrations attributed to RWC in winter were up to 10- or 15-fold, compared to the corresponding concentrations in summer. During the considered six-yearly period, the spatially highest predicted fractions of RWC of the annual PM$_{2.5}$ concentrations ranged from 12 to 14 %. In winter, the corresponding contributions ranged from 16 to 21 %. The RWC contribution was higher than the corresponding urban vehicular traffic contribution at all the residential stations during all years. The study has highlighted new research needs for the future, in particular (i) the modelling of the RWC emissions that

would be explicitly based on the actual ambient temperatures, and (ii) the modelling of the impacts of the most important holiday periods on the emissions from RWC.

## 1 Introduction

Exposure to fine particulate matter (PM$_{2.5}$) in ambient air has been shown to be associated with adverse health effects, such as acute lower respiratory infections, asthma, chronic obstructive pulmonary disease, lung cancer and cardiovascular disease,

as well as excess mortality (e.g., Horne et al., 2018; Anenberg et al., 2018; Li et al., 2016; Pope and Dockery, 2006; Pope et





al., 2020; Lelieved et al., 2020). It has been estimated that 4.2 million deaths per year have been caused globally by ambient PM$_{2.5}$ concentrations (Cohen et al., 2017).

One of the major sources of PM$_{2.5}$ concentrations is residential wood combustion (RWC). Wood combustion is a significant source of energy for cooking and heating in many countries worldwide (e.g., Bonjour et al., 2013; WHO, 2016). Also in
developed countries, where RWC has largely been replaced by other forms of energy in the 20$^{th}$ century, wood is still widely used as a heating source, especially in rural areas (Fernandes et al., 2007). In urban areas, wood-burning stoves and fireplaces are commonly used as supplementary heating method and for recreational use (WHO, 2015; Amann et al., 2018; Kukkonen et al., 2020).

Residential combustion has been estimated to cause 45 % of global anthropogenic, and 27 % of global total PM$_{2.5}$ emissions
in 2010, with an especially high share in Africa and parts of Asia (Klimont et al., 2017). In EU countries, residential combustion was estimated to account for 46 % of anthropogenic PM$_{2.5}$ emissions in 2005, of this 80 % was estimated to originate from the combustion of biomass (Amann et al., 2018). However, there was a substantial variation within the EU countries: the contribution of biomass combustion to PM$_{2.5}$ emissions ranged from less than 10 % in the Netherlands, Ireland, Cyprus, and Malta, to 70–80 % in Croatia, Latvia, and Lithuania.

In most Nordic and Baltic countries (except for Iceland), wood has commonly been used as a fuel for decades. Many of these countries have historically had ample resources of local wood (Denier van der Gon et al., 2015; Kukkonen et al., 2020). In many urban areas in the Nordic countries (i.e., Finland, Denmark, Norway, and Sweden), the RWC share of the local emissions of PM$_{2.5}$ has commonly been considerably lower than that in rural areas. This has been partly due to (i) the use of firewood only as a supplementary heating method in cities, and (ii) clearly, the other urban emission sources. However, the
total amount of RWC emissions has nevertheless been significant in part of the Nordic cities, either due to many RWC-heated detached houses, or the heating of larger blocks of flats by RWC (Kukkonen et al., 2020). For instance, in the Helsinki Metropolitan Area (HMA), RWC has been estimated to have caused 45 % of the urban PM$_{2.5}$ emissions from combustion in 2021 (Korhonen et al., 2022).

From 1990 to 2015, residential biomass consumption increased in EU (Bertelsen and Mathiesen, 2020). According to Viana
et al. (2015), in some countries, such as Norway, Austria, Denmark, and Bulgaria, the increase continued from 1990 to 2012. In some other countries, e.g., Spain, Greece, Hungary, and Croatia, the contribution of biomass combustion decreased from 1990 to 2005, after which the contribution started to increase. The latter trend was partly due to the economic situation in these countries and the domestic supply of wood, and partly to climate policies. Substituting fossil fuels by renewable energy has been recommended by the EU. The political target has been set to increase the share of renewable energy to 32 % by the
year 2030 (EU Renewable Energy Directive (RED II) 2018/2001). The growing concern for the health impacts associated with the emissions attributed to RWC has resulted in a consideration of abatement measures for RWC. In the long run, this might possibly result in a decreasing trend in RWC, but due to the current requirements to decrease the use of fossil energy in the EU, a significant change is not expected in the near future.



The PM$_{2.5}$ emissions originating from RWC are commonly released at low altitudes and may therefore have a substantial
adverse impact on local air quality and human health. It is important to quantitatively determine the contribution of RWC
emissions to the PM$_{2.5}$ concentrations in ambient air and evaluate the impact of potential emission reduction measures.
Karagulian et al. (2015) evaluated in a global review that an average of 20 % of urban ambient PM$_{2.5}$ was originating from
domestic fuel burning; the corresponding value for vehicular traffic was 25 %. However, they also found that there was a
wide variation between different regions. In western, northwestern, and central and eastern European countries, these
contributions were 15 %, 22 % and 32 %, respectively. According to a review of source apportionment studies by Belis et al.
(2013), the relative contribution of biomass combustion to PM$_{2.5}$ concentrations in Europe was 15±7 %, the highest relative
contributions in urban areas occurring in the Alps and in northern Europe.

In countries where wood is used for residential heating, the seasonal variation of RWC is pronounced (Klimont et al., 2017).
Episodes with high PM$_{2.5}$ concentrations often occur in winter, due to both intensive local emissions, and meteorologically
stable periods and low wind speeds. Trompetter et al. (2010) have compared PM$_{2.5}$ source contributions in winter and
summer, based on several source apportionment studies in New Zealand; the contribution of wood combustion ranged from
63 % to 91 % in winter and from 9 % to 45 % in summer. In northern Italy, the contributions of biomass combustion in the
period including autumn and winter were 25–30 % and 27–31 % in urban and rural sites, respectively. In summer, the
corresponding contributions were 1 % and 3 % (Perrone et al., 2012).

In the HMA, Hellén et al. (2017) have previously evaluated the significance of RWC to the concentrations of benzo(a)pyrene
(BaP), which is a good indicator substance for wood burning. Soares et al. (2014) and Aarnio et al. (2016) evaluated in detail
the contribution of different emission sources on the PM$_{2.5}$ concentrations in the HMA; however, these earlier studies did not
include the contributions originating from wood combustion. The first attempts to estimate the effect of wood combustion on
the PM$_{2.5}$ concentrations in the HMA were described by Ahtoniemi et al. (2010). More recently, Teinilä et al. (2022) have
assessed the impact of residential combustion on air quality in a detached housing area in Helsinki, based on air quality
measurements.

An unprecedentedly detailed emission inventory for RWC has been compiled in the HMA by the Helsinki Region
Environmental Services Authority (Kaski et al., 2016). Kukkonen et al. (2020) used this emission inventory to model the
PM$_{2.5}$ concentrations, including RWC. This study also evaluated the concentrations of PM$_{2.5}$ and the related contributions of
RWC in three other Nordic cities, viz. Copenhagen, Oslo, and Umeå. Kukkonen et al. (2020) focused on estimating the
annually averaged concentrations, and the study also addressed solely the PM$_{2.5}$ concentrations during a single year, for each
of the selected target cities. Orru et al. (2022) studied also the health impacts of PM$_{2.5}$ from RWC in the same four Nordic
cities. In another study, Kukkonen et al. (2018) evaluated the contributions of various source categories to the concentrations
of PM$_{2.5}$ in the HMA during a multidecadal period. This study also addressed solely annual average concentrations. The
above-mentioned studies included only a fairly limited evaluation of the model predictions against measured data; these
previous studies also did not include any in-depth analysis of the model performance or diagnostic model evaluation.



The main aim of this article is to investigate in-depth the contribution of RWC to the PM$_{2.5}$ concentrations in the HMA for several years (2009-2014). The specific objectives were (i) to evaluate the seasonal and shorter-term variations of the concentrations attributed to RWC, including their spatial variability, (ii) to analyze both the temporal and spatial year-to-year

variation of pollution from RWC, and (iii) to evaluate the model predictions against data in more depth, including diagnostic evaluation. The results can be used for understanding better both the inter-annual and shorter-term temporal variations of the pollution from RWC. The results can also be used for developing more effective policies for the abatement of pollution attributed to RWC, and for deriving insights for an improved modelling of the contributions from RWC.

## 2 Methods

### 2.1 The considered domain and the measurement network for concentrations

### 2.1.1 Modelling domain

The Helsinki Metropolitan Area (HMA) is an agglomeration of four cities: Helsinki, Espoo, Vantaa, and Kauniainen. The total population of this agglomeration is approximately 1.21 million (2022). The cities are in a fairly flat coastal area by the Baltic Sea. The annual average temperature in the centre of Helsinki is currently 6.5 °C; the monthly average ranges from -

3.8 °C in February to 18.1 °C in July (Jokinen et al., 2021). The locations of the cities and the measurement stations selected for this study are presented in Fig. 1.

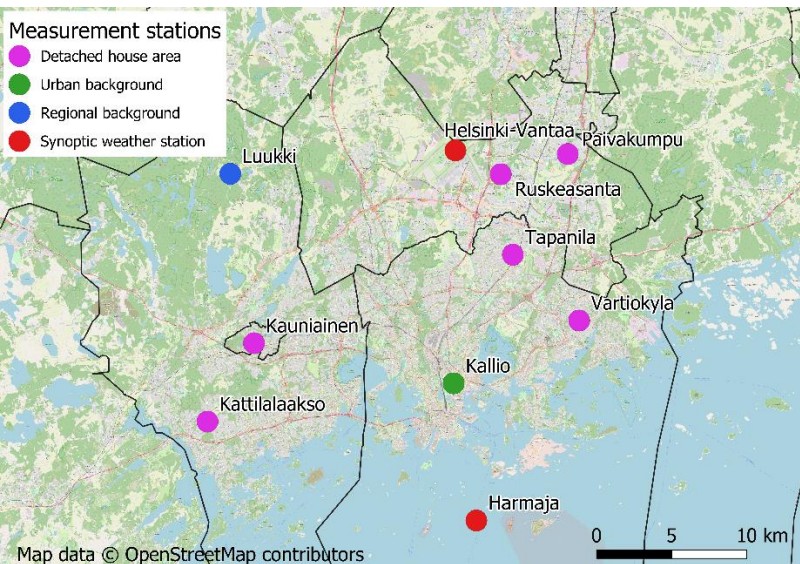

**Figure 1: The cities in the Helsinki Metropolitan Area and the air quality measurement stations used for this study. Three**
**categories of air quality stations were included: detached house areas, and urban and regional backgrounds. The urban background station is located in central Helsinki and the regional background station is in the northern part of the city of Espoo.**





**We also used weather data from two synoptic meteorological stations. © OpenStreetMap contributors 2019. Distributed under the Open Data Commons Open Database License (ODbL) v1.0.**

### 2.1.2 Concentration measurement network

We have used the hourly time series of the PM$_{2.5}$ concentrations, measured at eight air quality measurement stations operated by the Helsinki Region Environmental Services Authority (HSY). The locations of the stations are presented in Fig. 1. The classification of the sites, the years of measurement and the measurement devices are presented in Table 1. The monitoring height for all these sites was approximately 4 m.

**Table 1. The measurement sites used in this study, their site classifications, the years of measurement and the measurement devices for PM$_{2.5}$. The detached house areas have been additionally classified as more densely or more sparsely built ones.**

| Name of the site | Site classification | | Measurement year(s) | Measurement device(s) |
|---|---|---|---|---|
| Päiväkumpu | Detached house area | Densely built | 2011 | FH 62 I-R |
| Tapanila | Detached house area | Densely built | 2013 | FH62 I-R |
| Ruskeasanta | Detached house area | Densely built | 2014 | FH62 I-R |
| Vartiokylä | Detached house area | Sparsely built | 2009–2013 | Grimm 180 |
| | | | 2013–2014 | SHARP 5030 |
| Kattilalaakso | Detached house area | Sparsely built | 2012 | SHARP 5030 |
| Kauniainen | Detached house area | Sparsely built | 2013 | SHARP 5030 |
| Kallio | Urban background | | 2009–2014 | TEOM 1400 AB |
| Luukki | Regional background | | 2009 | FH 62 I-R, TEOM 1400 AB |
| | | | 2010 | FH 62 I-R |
| | | | 2011–2012 | FH 62 I-R, SHARP 5030 |
| | | | 2013–2014 | FH 62 I-R |

We have selected six measurement stations located in detached house areas (Vartiokylä, Päiväkumpu, Kattilalaakso, Kauniainen, Tapanila, and Ruskeasanta). The residential wood combustion emissions in this region are almost exclusively

originating from residential combustion in detached houses (Kukkonen et al., 2020). The urban and regional background stations at Kallio and Luukki, respectively, have been selected for the analysis of urban and local concentration increments.





The concentrations were measured with continuous measurement methods. FH 62 I-R monitor is based on the attenuation of β-rays by a filter, which is directly related to the amount of mass on the filter. Grimm 180 instrument uses an optical technique, in which particles are divided into different sizes in diameter based on light scattering. SHARP 5030

(Synchronized Hybrid Ambient Real-time Particulate Monitor) is a particulate monitor combining light scattering photometry and beta radiation attenuation. TEOM 1400 AB (Tapered Element Oscillating Microbalance) uses the tapered element oscillating microbalance technique to measure the mass concentration of particulate matter on a filter (Walden et al., 2010). The measured values were corrected with calibration equations based on Walden et al. (2010); these ensure equivalence with reference methods.

The surroundings of the measurement stations in detached house areas are substantially different. The immediate vicinity of stations of three sites (Päiväkumpu, Tapanila, and Ruskeasanta) is more densely built, compared to that of the other three considered stations in detached house areas. The site of Kattilalaakso is located lower than its surroundings, which may be unfavourable for the efficient mixing of pollution. All stations in detached house areas are located in regions with relatively low traffic volumes, except for the site of Ruskeasanta, as there is a densely trafficked highway at a distance of 700 m from

the site. This highway had an estimated average traffic volume on weekdays of 58 200 vehicles per day in 2013 (Malkki and Loukkola, 2015).

The urban background station at Kallio is located on the edge of a sports field in the city centre, at a distance of 80 m from a street with an average traffic volume on weekdays of 6 300–8 800 vehicles per day (2009–2014). The regional measurement station of Luukki is located in a rural area. However, it may occasionally be exposed to pollution caused by vehicular traffic

on a local minor road, and previously also by a camping centre, which was situated in the vicinity of the site, until the station has been moved a distance of 300 m further from this local source in May 2012.

## 2.2 Emission inventories

We have evaluated the $PM_{2.5}$ emissions from urban RWC and vehicular traffic. It has previously been found that the contribution of other urban source categories to the $PM_{2.5}$ concentrations has not been significant in this region. For shipping

and harbour activities, the contribution in the three-year period 2012–2014 has been estimated to exceed 10 % only in the immediate vicinity of major harbours (Kukkonen et al., 2018). The contribution of power production to local $PM_{2.5}$ concentrations has been negligible in most parts of the area (Hannuniemi et al., 2016).

Evaluation of the emissions originating from RWC and traffic emissions is presented briefly in this paper. For a more detailed description of the methodology, the reader is referred to Kukkonen et al. (2018).

### 2.2.1 Evaluation of the emissions from residential wood combustion

The emissions of RWC used in this study were based on an emission inventory by the local environmental authority, HSY, for 2013–2014. The amount of wood combusted in different types of fireplaces, and the habits for wood combustion were estimated using a questionnaire (Kaski et al., 2016). The results were applied to all detached and semi-detached houses in the



regional register for dwellings of the HMA. The emission factors for different types of fireplaces were based on the results of
a national measurement program and available literature (Kaski et al., 2016; Savolahti et al., 2016).

The meteorological variables, especially ambient temperature, influence the amount of RWC. The average variations of RWC between months, days of the week and hours of the day were included in the model, using coefficients based on the temporal distributions. These coefficients were based both on the questionnaire for 2013–2014 and on a previous survey for HMA, in which the temporal variation of fireplace usage was estimated in detail for 2008–2009 (Gröndahl et al., 2012;
Kaski et al., 2016). However, the impact of the actual temporal variation of ambient temperatures (e.g., based on measured temperature data for each hour) has not been explicitly considered, due to the lack of sufficiently detailed data.

The same spatial distribution of the emissions from RWC was used in the modelling for all the considered years. However, an estimate of the variation of total annual emission value was computed based on the available data on the number of detached houses, firewood consumption and the temporal changes in the share of different heating methods. This variation is
described in detail by Kukkonen et al. (2018). The annual total emissions in the HMA from RWC in 2009–2014 ranged from 189 to 175 t/a.

The emissions were evaluated separately for three different source categories: heating boilers, sauna stoves, and other fireplaces. The temporal variation is different for each of these categories: sauna stoves are used throughout the year, whereas the other fireplace types are mostly used during cold seasons. The seasonal variation is described in detail in
Appendix A.

The emission height, including the initial plume rise, was assumed to be equal to 7.5 m. This value is based on the average height of the detached and semi-detached houses in the area, and an estimated plume rise for the applied stove techniques.

In the HMA, 90 % of the detached houses have wood combustion appliances. However, only 2 % of the houses use RWC as a primary heating method.

**2.2.2 Evaluation of the emissions from vehicular traffic**

The traffic emissions were evaluated for vehicular exhaust and suspension for the roads and streets in the HMA for 2009–2014. Traffic emissions were evaluated for 26 536 line sources.

The spatial distribution of traffic volume data was computed using the EMME/2 transportation planning system for the year 2008 (HSL, 2011); this data was provided by the Helsinki Region Transport. The data consisted of mileage for three selected
hours for a day for each road link and regression-based factors for evaluating hourly traffic volumes. These data were given for weekdays, Saturdays, and Sundays. Exhaust emissions for each year were calculated using average emission factors and total emission values for HMA based on a national calculation system for traffic emissions, called LIPASTO (Mäkelä and Auvinen, 2009). The annual total emissions in the HMA originating from exhausts have a decreasing trend, these ranged from 214 t/a in 2009 to 124 t/a in 2014.

Suspension emissions were evaluated from exhaust emissions with a semi-empirical modelling approach, based on the average monthly ratio of concentrations from suspension and exhaust emissions. These coefficients were computed using





previous concentration results computed with emissions from a detailed road dust suspension model FORE (Kauhaniemi et al., 2011, 2014). The annual total emissions in the HMA originating from suspension ranged from 85 to 83 t/a from 2009 to 2014.

## 2.3 Meteorological measurements and modelling

We used the synoptic weather and radiation observations from Helsinki-Vantaa airport, located 18 km north of the Helsinki city centre, synoptic weather observations from the marine station of Harmaja, located on an island 7 km south of the city centre, and sounding observations from Jokioinen, 90 km northwest of Helsinki, for 2009–2014. The locations of the synoptic stations are presented in Fig. 1.

The meteorological pre-processing model MPP-FMI (Karppinen et al., 2000a) was used for analyzing the measured meteorological data. The model is based on the energy budget method of van Ulden and Holtslag (1985). The output of the MPP-FMI includes an hourly time series of meteorological data needed for the dispersion modelling, such as temperature, wind speed, wind direction, calculated atmospheric turbulence parameters, and the boundary layer height. The same meteorological data was applied to the whole HMA.

## 2.4 Atmospheric dispersion modelling

Urban scale dispersion of emissions from RWC and traffic were evaluated with multiple-source Gaussian dispersion models. The dispersion parameters were modelled as a function of Monin–Obukhov length, friction velocity, and boundary layer height. Fine particulate matter ($PM_{2.5}$) was treated as an inert substance; chemical reactions or aerosol transformation were not included in the modelling. The influence of terrain was included in the model as average surface roughness. Time series of hourly concentrations of $PM_{2.5}$ were computed for the HMA for 2009–2014.

The dispersion of RWC emissions was computed with the Urban Dispersion Model of the Finnish Meteorological Institute UDM-FMI (Karppinen et al., 2000b), which is a multiple source Gaussian dispersion model for point, area, and volume sources. The model has been evaluated against measured data, e.g., by Karppinen et al. (2000c). In the model computations, the RWC emissions were treated as area emissions uniformly distributed in squares of size 100 m x 100 m.

The dispersion of vehicular emissions was evaluated with the CAR-FMI model (Contaminants in the Air from a Road – Finnish Meteorological Institute), which is a Gaussian finite-length line source model (e.g., Härkönen, 2002; Karppinen et al., 2000b; Kukkonen et al., 2001). The CAR-FMI model has been evaluated against measured data, e.g., by Karppinen et al. (2000c), Kauhaniemi et al. (2008), Aarnio et al. (2016), Singh et al. (2014) and Srimath et al. (2017). Street canyon dispersion modelling was not applied. The coefficients of the variation of weekly emissions were included in the calculations.

The calculation grid consisted of two sub-grids: the sub-grid for RWC, with a horizontal resolution of 100 m x 100 m, and the sub-grid for traffic, including 52 301 calculation points, with spatial resolution ranging from 20 m in the vicinity of the roads to 500 m in background areas.





The regional background concentrations were based on concentrations computed with a global-to-meso-scale dispersion
model SILAM (Sofiev et al., 2006, 2015). The concentrations were evaluated for the European domain, and their
computation has been described in detail by Kukkonen et al. (2018). For estimating the hourly average regional background
concentrations, we selected four SILAM grid points closest to the HMA and calculated the average of the concentrations at
these four locations for all the chemical components of $PM_{2.5}$, except for mineral dust. For mineral dust, we used the
minimum concentration value in the four selected points.

**2.5 Statistical parameters**

For evaluating model performance, we have computed the following statistical parameters: the index of agreement (IA), the
square of the correlation coefficient, also called R-squared (R2), the normalized mean square error (NMSE), the factor-of-
two (F2), and the fractional bias (FB).

The index of agreement is defined as (Willmott, 1981)

$$IA = 1 - \frac{\overline{(C_P - C_O)^2}}{\overline{(|C_P - \overline{C_O}| + |C_O - \overline{C_O}|)^2}} , \qquad (1)$$

where $C_P$ and $C_O$ are the predicted and observed concentration, and overbar denotes the average over the dataset. The index
of agreement is a measure of the degree to which observed deviations about $\overline{C_O}$ correspond to predicted deviations about $\overline{C_O}$,
and it is sensitive to the differences between the observed and predicted means (Willmott, 1981). The index of agreement
varies from 0.0 to 1.0; the latter value corresponds to a perfect agreement. The value of the IA of approximately 0.4
corresponds to the agreement of two random time series, which have the same average value (Karppinen et al., 2000c).

The parameters R2, NMSE, and F2 are measures of correlation of the predicted and observed concentration time series. R2 is
the square of the correlation coefficient R:

$$R = \frac{\overline{(C_O - \overline{C_O})(C_P - \overline{C_P})}}{\sigma_{C_O} \sigma_{C_P}} , \qquad (2)$$

where $\sigma_C$ denotes standard deviation over the dataset. NMSE is defined as (e.g., Chang and Hanna, 2004)

$$NMSE = \frac{\overline{(C_O - C_P)^2}}{\overline{C_O} \, \overline{C_P}} . \qquad (3)$$

Factor-of-two is defined as the fraction of data for which $0.5 \leq C_P/C_O \leq 2$, i.e., it describes the share of predictions within a
factor of two compared with observations.

Fractional bias is a measure of the agreement of the observed and predicted mean concentrations, and it is defined as

$$FB = \frac{2(\overline{C_P} - \overline{C_O})}{\overline{C_P} + \overline{C_O}} . \qquad (4)$$

Values of FB between -0.67 and 0.67 describe under- and over-estimation by a factor of two.



## 3 Results and discussion

We first evaluate the predicted concentrations against measurements. Second, we present and discuss the predicted concentrations of $PM_{2.5}$, their year-to-year variation, and the difference between concentration values annually and in winter. We also compare the contributions from RWC and traffic to the $PM_{2.5}$ concentrations in residential areas, and examine the

temporal variations of these contributions.

We have also presented an analysis of the representativity of the measurement stations in areas containing detached houses in Appendix B. This analysis was done in terms of the location and amount of emissions from RWC in the vicinity of each measurement station. The stations were exposed to substantially varying amounts of emissions originating from RWC.

### 3.1 Evaluation of predicted concentrations against measurements

**3.1.1 Statistical analysis**

The annual average concentrations and statistical parameters for the selected measurement stations are presented in Table 2. Statistical parameters were calculated from daily average concentrations. The concentrations were measured for all the considered years only at one residential site (Vartiokylä), and for the regional and urban background sites.

**Table 2. Observed and predicted annual average concentrations (μg/m³) and statistical parameters (IA, R2, NMSE, FB, F2) for six measurement stations in residential areas, and for the urban and regional background stations at Kallio and Luukki, respectively. Parameters were calculated from daily concentrations. The number of daily averages (N) is also presented.**

|  |  | 2009 | 2010 | 2011 | 2012 | 2013 | 2014 |
|---|---|---|---|---|---|---|---|
| Vartiokylä | $\overline{C_O}$ | 7.4 | 8.1 | 7.4 | 6.6 | 6.8 | 9.8 |
|  | $\overline{C_P}$ | 8.2 | 8.4 | 7.5 | 7.4 | 7.5 | 9.0 |
|  | IA | 0.75 | 0.74 | 0.81 | 0.75 | 0.69 | 0.72 |
|  | R2 | 0.39 | 0.32 | 0.48 | 0.39 | 0.30 | 0.32 |
|  | NMSE | 0.41 | 0.49 | 0.46 | 0.57 | 0.50 | 0.55 |
|  | F2 | 76 | 73 | 74 | 73 | 68 | 65 |
|  | FB | 0.10 | 0.04 | 0.02 | 0.11 | 0.09 | -0.08 |
|  | N | 342 | 362 | 358 | 365 | 351 | 365 |
|  |  |  |  |  |  |  |  |
| Päiväkumpu | $\overline{C_O}$ |  |  | 10.8 |  |  |  |
|  | $\overline{C_P}$ |  |  | 7.9 |  |  |  |
|  | IA |  |  | 0.76 |  |  |  |
|  | R2 |  |  | 0.39 |  |  |  |



|  |  |  |  |  |
|---|---|---|---|---|
|  | NMSE | 0.55 |  |  |
|  | F2 | 56 |  |  |
|  | FB | -0.31 |  |  |
|  | N | 362 |  |  |
|  |  |  |  |  |
| Kattilalaakso | $\overline{C_O}$ |  | 8.2 |  |
|  | $\overline{C_P}$ |  | 7.5 |  |
|  | IA |  | 0.70 |  |
|  | R2 |  | 0.29 |  |
|  | NMSE |  | 0.59 |  |
|  | F2 |  | 63 |  |
|  | FB |  | -0.10 |  |
|  | N |  | 361 |  |
|  |  |  |  |  |
| Kauniainen | $\overline{C_O}$ |  |  | 7.1 |
|  | $\overline{C_P}$ |  |  | 7.3 |
|  | IA |  |  | 0.65 |
|  | R2 |  |  | 0.24 |
|  | NMSE |  |  | 0.56 |
|  | F2 |  |  | 66 |
|  | FB |  |  | 0.02 |
|  | N |  |  | 360 |
|  |  |  |  |  |
| Tapanila | $\overline{C_O}$ |  |  | 9.1 |
|  | $\overline{C_P}$ |  |  | 7.8 |
|  | IA |  |  | 0.67 |
|  | R2 |  |  | 0.24 |
|  | NMSE |  |  | 0.49 |
|  | F2 |  |  | 65 |
|  | FB |  |  | -0.16 |
|  | N |  |  | 360 |
|  |  |  |  |  |
| Ruskeasanta | $\overline{C_O}$ |  |  | 11.3 |

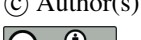



|  |  |  |  |  |  |  |  |
|---|---|---|---|---|---|---|---|
|  | $\overline{C_P}$ |  |  |  |  |  | 9.4 |
|  | IA |  |  |  |  |  | 0.69 |
|  | R2 |  |  |  |  |  | 0.26 |
|  | NMSE |  |  |  |  |  | 0.55 |
|  | F2 |  |  |  |  |  | 65 |
|  | FB |  |  |  |  |  | -0.18 |
|  | N |  |  |  |  |  | 359 |
| Kallio | $\overline{C_O}$ | 8.4 | 9.0 | 7.8 | 7.5 | 7.0 | 8.1 |
|  | $\overline{C_P}$ | 8.0 | 8.4 | 7.7 | 7.2 | 7.2 | 8.8 |
|  | IA | 0.75 | 0.65 | 0.75 | 0.69 | 0.64 | 0.72 |
|  | R2 | 0.37 | 0.19 | 0.37 | 0.29 | 0.22 | 0.36 |
|  | NMSE | 0.35 | 0.58 | 0.55 | 0.61 | 0.54 | 0.61 |
|  | F2 | 73 | 67 | 63 | 66 | 66 | 65 |
|  | FB | -0.06 | -0.07 | -0.02 | -0.05 | 0.03 | 0.08 |
|  | N | 365 | 363 | 358 | 363 | 364 | 365 |
| Luukki | $\overline{C_O}$ | 7.2 | 8.5 | 7.7 | 7.1 | 6.3 | 7.4 |
|  | $\overline{C_P}$ | 7.3 | 7.6 | 7.2 | 7.0 | 6.7 | 8.4 |
|  | IA | 0.72 | 0.63 | 0.65 | 0.68 | 0.59 | 0.65 |
|  | R2 | 0.33 | 0.15 | 0.20 | 0.26 | 0.15 | 0.29 |
|  | NMSE | 0.45 | 0.76 | 0.86 | 0.72 | 0.71 | 0.77 |
|  | F2 | 67 | 62 | 55 | 63 | 58 | 61 |
|  | FB | 0.01 | -0.11 | -0.07 | -0.02 | 0.05 | 0.13 |
|  | N | 365 | 360 | 358 | 348 | 365 | 364 |

The differences between the predicted and observed concentrations at RWC stations are partly due to the setup of the model.

The emissions of RWC were considered on a spatial resolution of 100 m x 100 m. Clearly, this can average out the spatial emission distribution on a finer scale. The highest measured short-term local concentration peaks may therefore be under-predicted by the model. These impacts are probably highest in the more densely built residential areas.

The predicted concentrations near the sources are also sensitive to the evaluation of the effective emission height. Clearly, the effective emission height is dependent also on the meteorological conditions, and the structural details of buildings in





each region. All the observed values were measured at a height of 4 m. Near an emission source the predicted concentration at the ground level may not accurately represent the observed concentration value at the height of 4 m.

The overall agreement of predicted concentrations with measurements can be considered to be good or fairly good for all stations.

In the regional background station, IA ranges from 0.59 to 0.72, and fractional bias from −0.11 to 0.13. For the urban

background station, the IA ranges from 0.64 to 0.75, and fractional bias from −0.07 to 0.08. The model agreement is therefore slightly better at the urban background station, compared with the regional background. The corresponding model performance statistics for the permanent residential site (Vartiokylä) are close to those at the urban background station (IA ranges from 0.69 to 0.81, and FB ranges from −0.08 to 0.11). For the other residential sites, the IA ranges from 0.65 to 0.76, which is close to the corresponding agreement at the urban background station. The FB values are different between more

densely (FB from −0.31 to −0.16) and more sparsely built areas (FB from −0.1 to 0.02); the model slightly under-predicts in densely built areas.

Regarding the permanent residential site, the year-to-year variation of model performance can be considered to be substantial. These differences are probably mainly caused by the changing weather conditions. The modelling of the amounts of RWC was based on semi-empirical temporal profiles of the amount of RWC (for months, days of the week and hours of

the day), instead of the actual measured temperature values.

According to our previous studies (e.g., Kukkonen et al., 2018), the model performance measures for traffic stations in the HMA were in the same range, compared with those found at the RWC stations in this study. This provides evidence that the impacts of RWC were modelled with a comparable accuracy, as the modelling of the impacts of vehicular traffic.

**3.1.2 Seasonal and monthly average concentrations**

We have computed average concentrations for the seasons of the year, during the whole of the considered six-year period. The seasons have been defined as follows: winter as January, February, and December, spring as March, April, and May, summer as June, July, and August, and autumn as September, October, and November. Seasonal average concentrations are presented in Figs. 2a–c for three categories of stations.



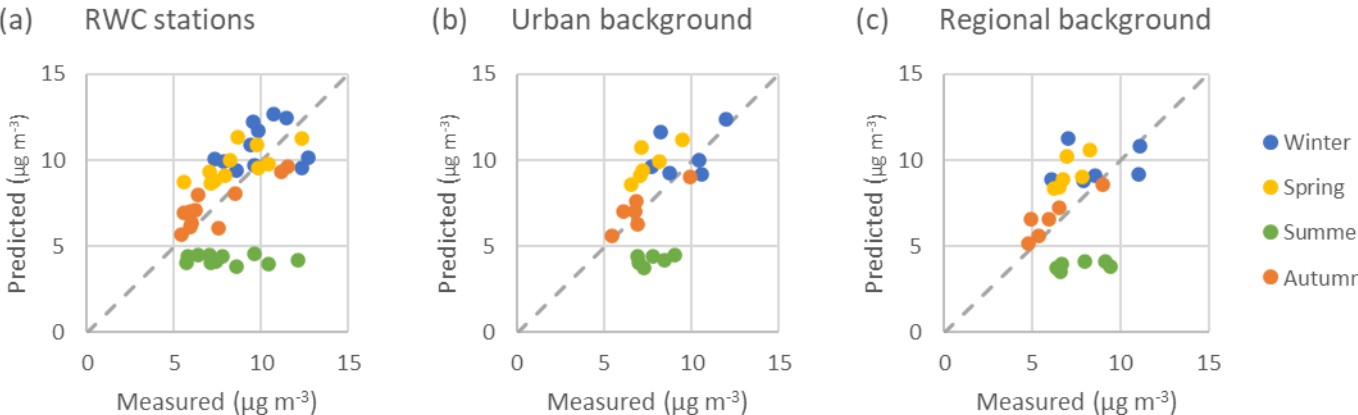

**Figure 2: Predicted and measured concentrations (μg/m³) during seasons of the year for three categories of stations, i.e., residential, and urban and regional background, during 2009–2014.**

In summer, the concentrations were substantially under-predicted for all the categories of stations. The regional background concentration values were extracted directly from the predictions of the chemical transport model SILAM. The model skill scores for the PM$_{2.5}$ concentrations of the SILAM model have previously been found to be relatively worse in summer, due to uncertain particulate matter components in summer, such as biogenic organics and the contributions from wild-land fires (Prank et al., 2016). The agreement in summer is worse for RWC stations than for the background stations, which may be due to some missing emission sources specifically in summer in the model, e.g., particulate matter from barbeques in some residential areas.

The agreement of predicted and measured concentrations was fairly good during the other seasons. There were over-predictions for most of the values in spring. In autumn, there were both over- and under-predictions. In winter, the concentrations at the residential stations were mostly slightly over-predicted, whereas there was no systematic over- or under-prediction at the regional background and urban stations. These over-predictions at the residential sites in winter were probably caused by the assumed semi-empirical seasonal variation of RWC emissions in winter; this variation function may not have been ideal for the meteorological conditions during the considered periods.

We have also analyzed the bias of the model predictions in terms of the severity of the winter. These results have been presented in Appendix C. These results showed that there were slightly more model over-predictions in case of the relatively warmer winters. The model under-predicted especially in December at the residential sites. This could have been caused by the increased recreational wood burning and cooking by wood-burning stoves during the Christmas holiday season, and in addition, the fireworks of the New Year celebration. The model under-predictions in December were not correlated with ambient temperatures. The results also highlight that the local winter holiday week in February has an impact on the amount of wood combustion.





### 3.1.3 Correlations of daily average concentrations and ambient temperatures

To analyze in-depth the impact of ambient temperature to local RWC concentrations in winter, we first selected the cases, for which the RWC contribution was above the average value, according to the model computations. The observed and predicted concentrations for these cases against the ambient temperatures are presented in Figs. 3a–m at the residential stations.







**Figure 3: Daily average measured (left-hand side panels) and predicted (right-hand side panels) concentrations as a function of the ambient temperature. The data has been presented only for the cases, in which the contribution attributed to RWC was above the average value at the residential stations. The linear trends of data in each figure have been presented by the blue lines. The overall average trend of observed concentrations on temperature for all years and stations is presented by the orange lines; this trend is therefore the same for all the panels on the left-hand side.**






Both the measured and predicted daily averaged concentrations show increasing trends towards the lower temperature values. The trends and the distribution of data points are similar for all the stations, both for the measured and predicted data. As expected, the observed concentrations were slightly better correlated with temperature than the predicted concentrations, with one exception (the residential site at Kattilalaakso).

In particular, at two residential stations (Tapanila and Kauniainen), there were high predicted daily concentrations at a moderate ambient temperature of −4 °C. We examined these cases in detail. The reason for predicting such high concentration values was connected to especially inefficient dilution of pollution during prevailing very low winds, combined with high predicted RWC emissions during a weekend. The observed concentrations at the time were not substantially elevated. However, we found that these days corresponded to the local winter holiday week, during which a

large fraction of people are traditionally travelling. The influence of the main holidays should therefore in principle be considered in modelling the temporal variation of RWC emissions.

## 3.2 Predicted spatial concentration distributions

The predicted spatial distributions of annual $PM_{2.5}$ concentrations, and those averaged for winter are presented in Figs. 4a–d for two years, 2009 and 2010. These concentrations include the contributions originating from both urban vehicular traffic

and RWC, and the regional background. We have chosen these two years to highlight the year-to-year variation of the RWC contributions. During the considered period (2009–2014), the concentrations of $PM_{2.5}$ in winter originated from urban wood combustion were the lowest in 2009 and the highest in 2010.



PM₂.₅ annual all 2009          PM₂.₅ annual all 2010

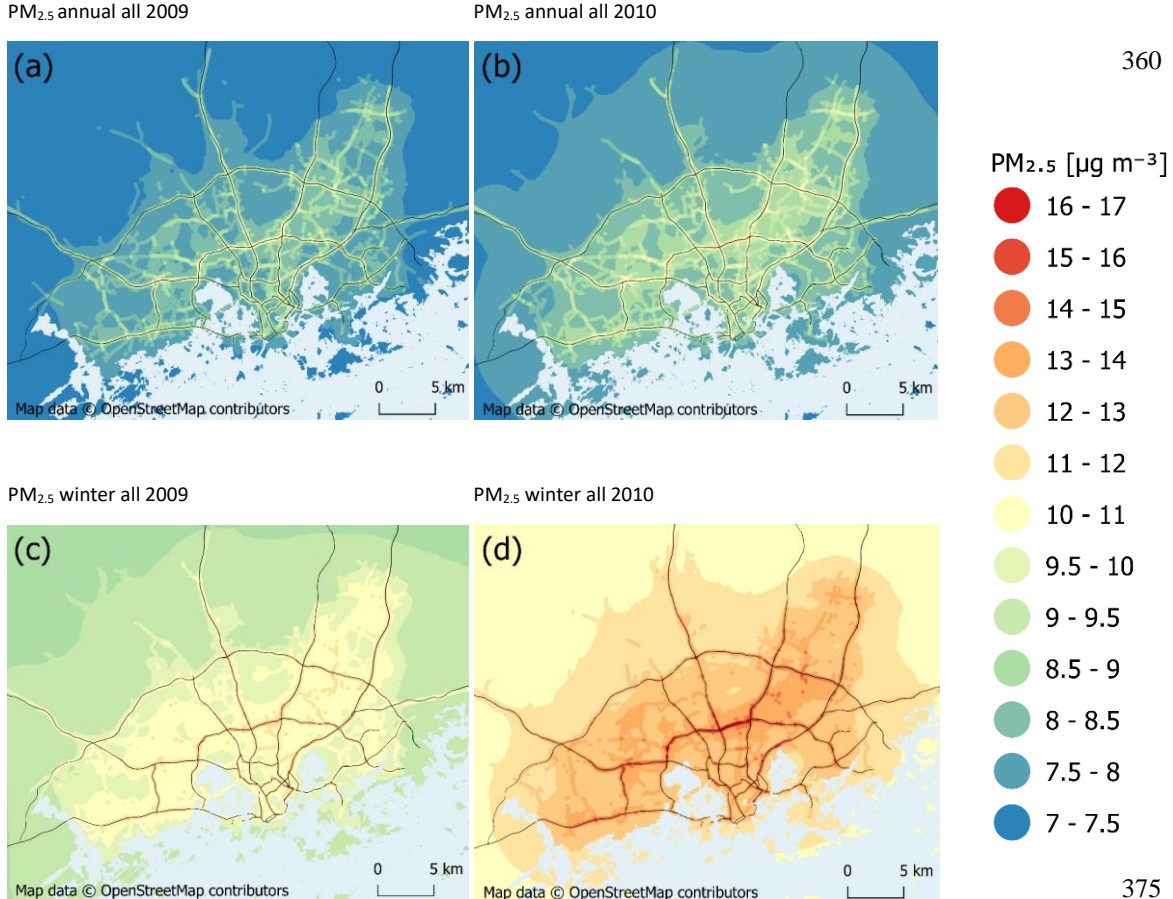



**Figure 4: The predicted average concentrations of PM₂.₅ (µg/m³) in the Helsinki region, as annual averages in 2009 (a) and 2010 (b), and as winter averages in 2009 (c) and 2010 (d). The main street and road network is presented with the black lines, and the sea areas in light blue colour. The physical scales of figures are indicated by the bars in the lower right-hand sides of each panel. © OpenStreetMap contributors 2019. Distributed under the Open Data Commons Open Database License (ODbL) v1.0.**


The centre of Helsinki is located on a peninsula, in the southern middle region of the maps. The highest annually averaged concentrations (Figs. 4a–b) occurred in the vicinity of major roads and streets, and in the suburban residential areas, to the north-west, north and north-east of the city centre. In city centre, the source contribution of RWC was low, due to almost negligible local emissions. The predicted annual average concentrations ranged from 7.2 to 11.0 µg/m³ and from 7.4 to 11.7

µg/m³ in 2009 and 2010, respectively.

The modelling takes into account the seasonal variation of the RWC emissions using a semi-empirical variation function, which has been assumed to be the same for all the years. The differences of the predicted annual concentrations in different





years are therefore caused by (i) the different urban meteorological conditions in different years (affecting the atmospheric dispersion), and (ii) by the different regional background concentrations.

In winter (Figs. 4c–d), the corresponding average concentrations were higher for both years, and these were more focused on the residential areas. The differences of the annual and winter concentrations in each year are caused by a clear difference between background concentrations but also by the higher RWC contribution in winter. The concentrations in winter ranged from 8.9 to 13.1 µg/m$^3$ in 2009 and from 10.5 to 16.4 µg/m$^3$ in 2010.

In winter, the highest concentrations from RWC to PM$_{2.5}$ were 1.7 and 2.8 µg/m$^3$ in 2009 and 2010, respectively. In summer,
the corresponding seasonal values were approximately 0.2 µg/m$^3$ with a negligible variation between different years. Consequently, the average concentrations attributed to RWC in winter were up to 10- or 15-fold, compared to corresponding concentrations in summer in 2009 and 2010, respectively.

The relative fractions of the contribution of RWC to the PM$_{2.5}$ concentrations is presented in Figs. 5a–d, as annual averages and in winter. The highest predicted fractions of RWC of the annual PM$_{2.5}$ concentrations were 12 % and 14 % in 2009 and
2010, respectively. In winter, the highest average contributions were 16 % in 2009 and 21 % in 2010.

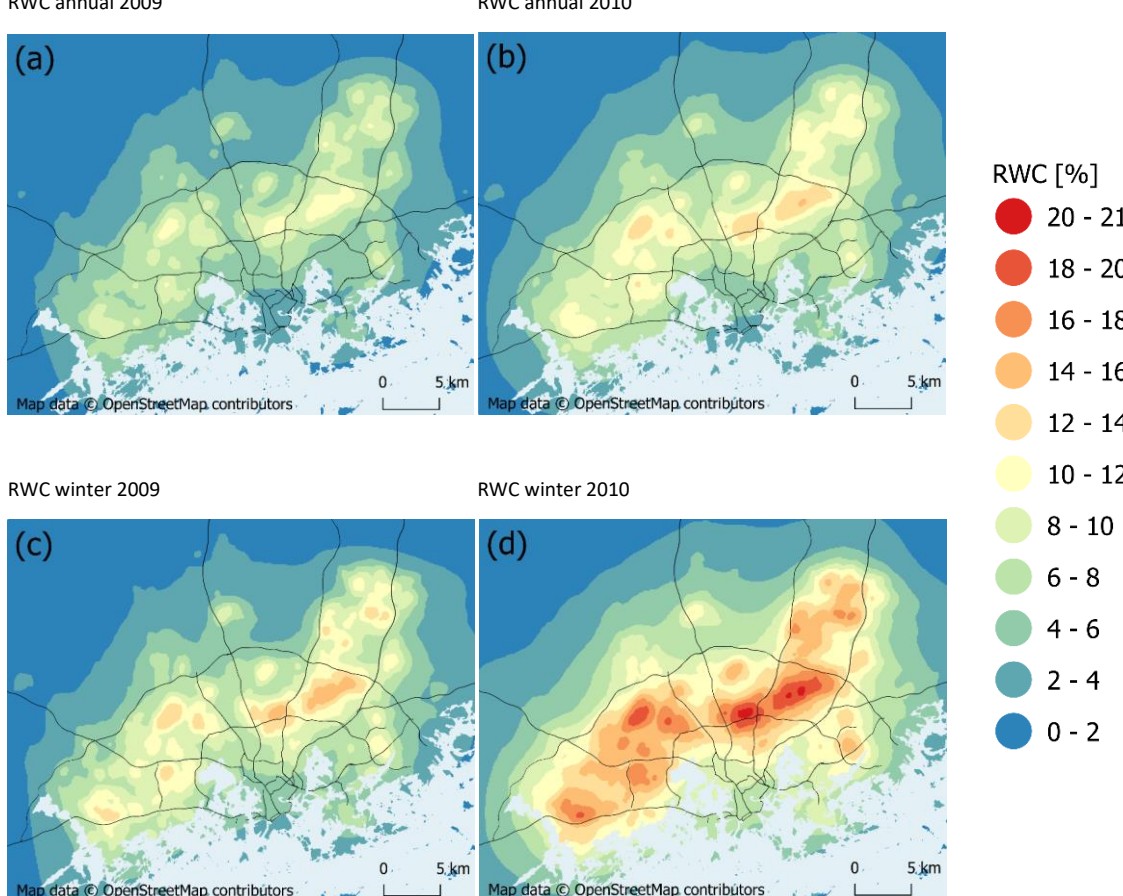





**Figure 5: Contribution of residential wood combustion to the concentrations of PM$_{2.5}$ annually (panels a–b) and in winter (panels c–d) in 2009 and 2010 (%). © OpenStreetMap contributors 2019. Distributed under the Open Data Commons Open Database License (ODbL) v1.0.**

## 3.3 Monthly average contributions from traffic and residential wood combustion

Predicted monthly concentrations, attributed to urban RWC and vehicular traffic are presented for measurement stations in Figs. 6a–c.





(a) Vartiokylä

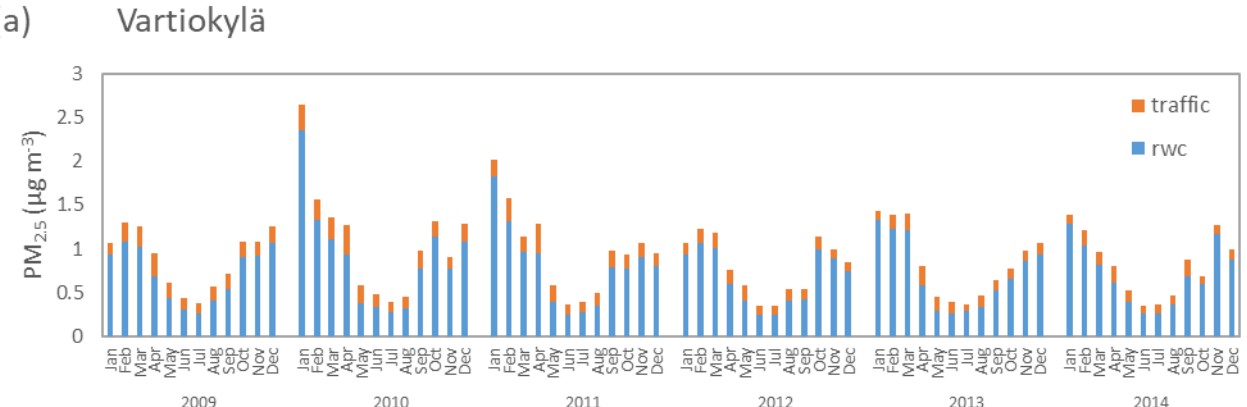

(b) Other RWC stations

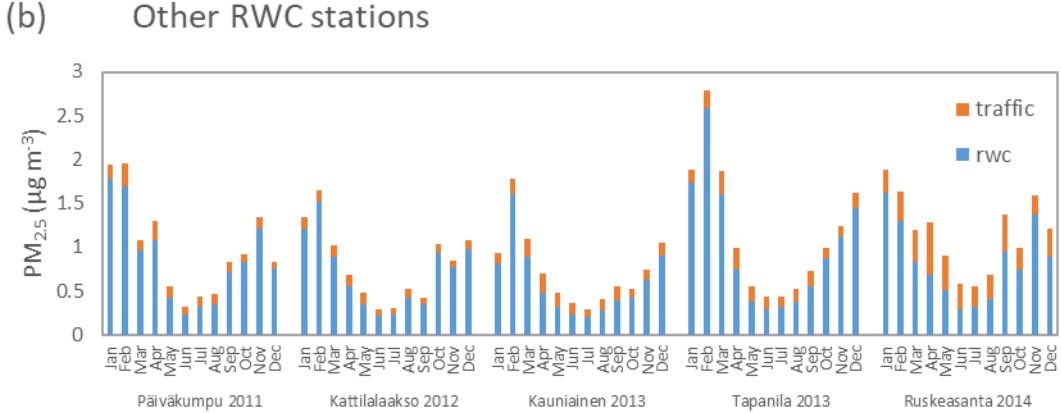

(c) Urban background

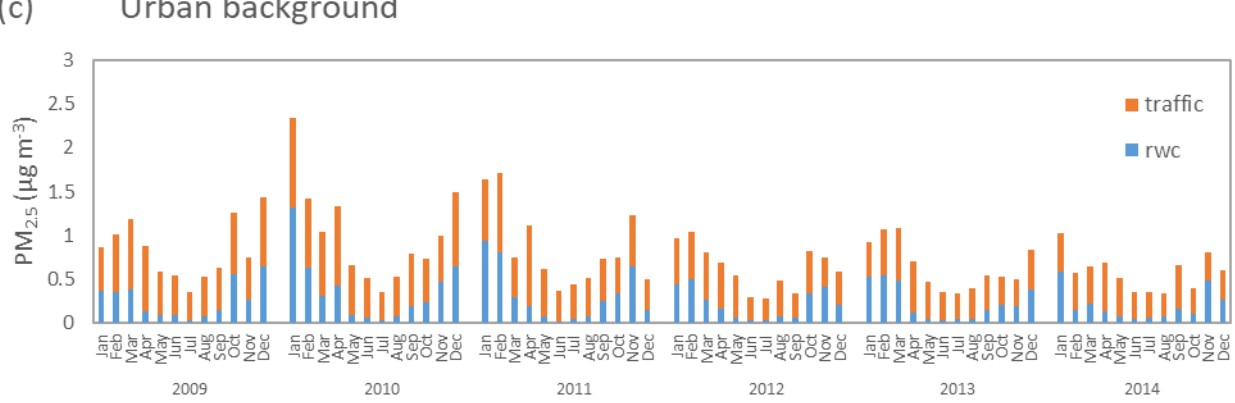

**Figure 6: Predicted monthly average concentrations originating from urban sources, attributed to urban vehicular traffic and residential wood combustion. The regional background has been excluded.**






Clearly, the PM$_{2.5}$ concentrations attributed to RWC were clearly higher in winter. In case of vehicular traffic, the seasonal variation is moderate. The predicted time series for the permanent residential site (Vartiokylä) highlights the year-to-year variation, which is caused mainly by the differences in urban meteorological conditions.

At all the RWC stations during all years, the RWC contribution was higher than the corresponding vehicular traffic

contribution for all the considered months. However, at one site (Ruskeasanta), the monthly average concentrations from traffic and RWC were comparable in summer; this has been caused by the emissions from a heavy trafficked highway in the vicinity of the site. At the urban background station in Kallio, the concentrations attributed to urban traffic were dominant.

## 4 Conclusions

Residential wood combustion (RWC) has been found to be an important source of fine particulate matter worldwide, and its

relative significance will probably increase in the future. Previous studies have indicated that in the Helsinki Metropolitan Area (HMA), RWC has had a significant impact on PM$_{2.5}$ emissions (Kukkonen et al., 2018, 2020) and on public health (Orru et al., 2022) in the early 2010s. Kukkonen et al. (2018) evidenced that the emissions of PM$_{2.5}$ originated from small-scale combustion in the HMA have increased slightly in time from 1980s to the early 2010s. The relative share of RWC with respect to the total emissions in that area has substantially increased during that period. The reasons for this relative increase

were that both the emissions from local vehicular traffic and the long-range transported background of the PM$_{2.5}$ concentrations have decreased during that period; the former by a factor of five (Kukkonen et al., 2018). The recent development of RWC in the HMA can be evaluated with measured concentrations of benzo(a)pyrene (BaP), which is predominantly originating from wood burning. During the period considered in this study (2009 – 2014), the annual average of BaP concentration at Vartiokylä ranged from 0.5 to 0.7 ng m$^{-3}$, and decreased to the level of 0.4 ng m$^{-3}$ in the late 2010s.

Rising energy prices in 2022, however, increased household wood burning, and the annual average of BaP concentration was at a slightly higher level of 0.5 ng m$^{-3}$, which is the same level as in 2009 – 2010 (Korhonen et al., 2023). Currently, the RWC emissions can therefore be estimated to be at the same or slightly lower level as in the beginning of 2010s. It has been previously estimated that Finnish RWC emissions will remain at a constant level in the 2020s or decline only slightly (Ministry of the Environment, 2019). However, the recent deterioration of the economic situation and high energy prices

may favour wood burning and increase emissions. With the simultaneous reduction in vehicular traffic emissions, the relative importance of RWC as a local source of PM$_{2.5}$ may thus further increase in the future.

In the HMA, the impact of urban RWC emissions to the PM$_{2.5}$ concentration has previously been evaluated on annual level, and with respect to three other Nordic cities (Kukkonen et al., 2020). In this study, we have substantially extended the previous analyses, by investigating in-depth both the year-to-year and seasonal variations of the emissions from RWC and

the corresponding concentrations, based both on model computations and measurements.



The model performance against measured data was statistically evaluated at six residential monitoring stations, and regional and urban background stations, during six years. The overall agreement of predicted concentrations with measurements was good or fairly good for all stations and years. The model performance was on the average better for the residential and urban background sites, compared with the regional background station. This provides evidence that the modelling of the emissions

and dispersion of RWC was more accurate than the modelling of the regional and urban background concentrations. However, the interannual variation of the model performance was substantial, mainly caused by the changing weather conditions from year to year. The modelling of the emissions of RWC was based on semi-empirical temporal profiles, which were assumed to be the same for each year. The model performance measures for vehicular traffic stations in the Helsinki region were previously found to be in the same range (Kukkonen et al., 2018), compared with those found at the RWC

stations in this study.

We also analyzed the seasonal variation of the model performance. In particular, in winter, the concentrations at the residential stations were mostly slightly over-predicted, whereas there was no systematic over- or under-prediction at the regional background and urban stations. These over-predictions at the residential sites in winter were probably caused by the inaccuracies due to the assumed semi-empirical seasonal variation of the RWC emissions. The analysis regarding the

monthly variation of the model performance showed, in particular, that the model under-predicted in December at the residential sites. This was most likely caused by the traditional increased recreational wood burning and cooking by wood-burning stoves during the Christmas holiday season. The modelling did not allow for the specific influence on RWC of holiday seasons.

As expected, at the residential sites, both the measured and predicted daily averaged concentrations were substantially higher

at lower ambient temperature values. The higher concentrations were caused both by the increased RWC and the more frequent inefficient dispersion conditions during periods of prevailing low temperatures. These trends and the distribution of data points were similar for all the residential stations, both for the measured and predicted data. In particular, it was found that there were some substantial model over-predictions during a local winter holiday week. During that period, a large fraction of people has traditionally been travelling, which we have not taken into account in the modelling.

The highest predicted annually averaged concentrations in the Helsinki region occurred in the vicinity of major roads and streets, and in the suburban residential areas, to the north-west, north and north-east of the city centre. In city centre, the source contribution of RWC was low, due to almost negligible local emissions.

In winter, the concentrations were clearly higher than the annual average concentrations, and the spatial distributions were more focused on the residential areas. The average concentrations attributed to RWC in winter were up to 10- or 15-fold,

compared to corresponding concentrations in summer, in 2009 and 2010, respectively. The interannual variation of concentrations was considerable especially in winter.

The regional background is the largest contributor to the concentrations of $PM_{2.5}$ in HMA. During the considered six-yearly period, the spatially highest predicted fractions of RWC of the annual $PM_{2.5}$ concentrations ranged from 12 to 14 %. In winter, the corresponding contributions ranged from 16 to 21 %. At all the residential stations during all the considered





years, the RWC contribution was higher than the corresponding contribution attributed to vehicular traffic, for all months of the year. At the urban background station in Kallio, the concentrations attributed to urban traffic were dominant.

The study has highlighted some research needs for the future. It would be more accurate, if the modelling of the RWC emissions would be directly based on the actual hourly meteorological parameters, especially the ambient temperatures. Clearly, this would necessitate a development of new semi-empirical, temperature-dependent temporal profile functions,

regarding the variation of RWC emissions, on daily, weekly and seasonal basis. The impacts of the most important holiday periods should also be taken into account in the modelling of the emissions originating from RWC. For this aim, one would need a more detailed survey of human activities, including especially the possible increased recreational use of fireplaces, and an evaluation of the fraction of people travelling during holidays.

**Appendix A. The temporal variation of RWC emissions**

The temporal variation functions for RWC emissions applied in the model are presented in Figs. A1a–c. These functions were determined by analyzing the datasets based on questionnaires for the years 2008–2009 (Gröndahl et al., 2012) and for the years 2013–2014 (Kaski et al., 2016). However, the number of boilers in the questionnaire was not adequate for estimating the seasonal variation, and it was therefore estimated based on monthly heating degree days and estimates for the amount of energy required for heating hot water. The same variation functions have been applied for all years.




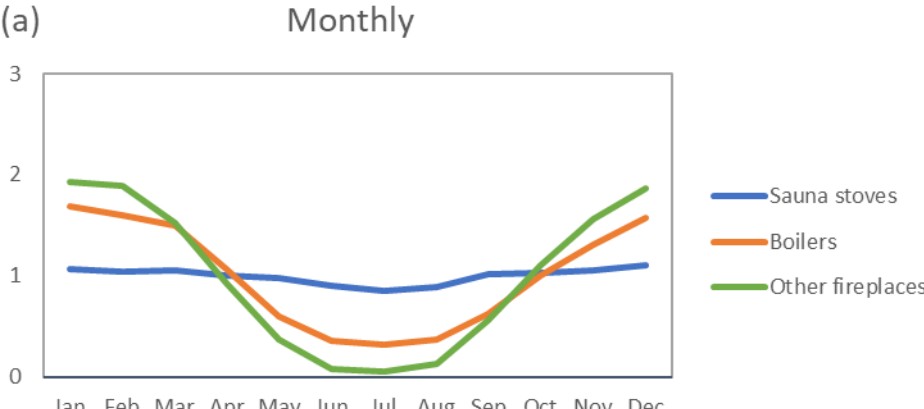

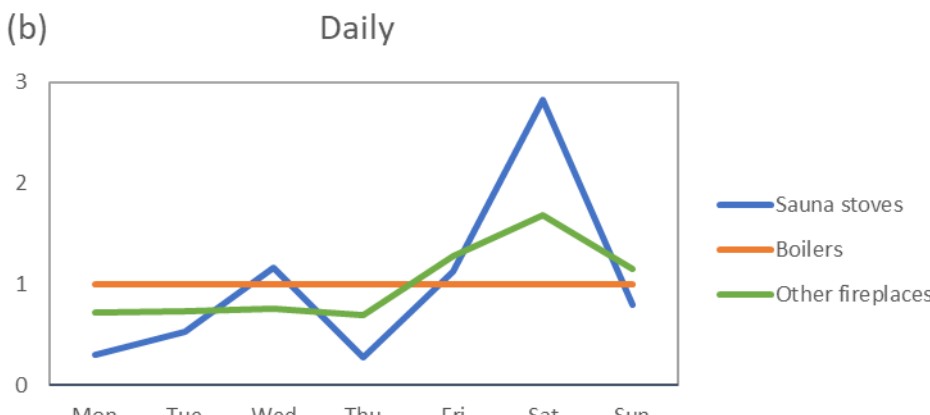

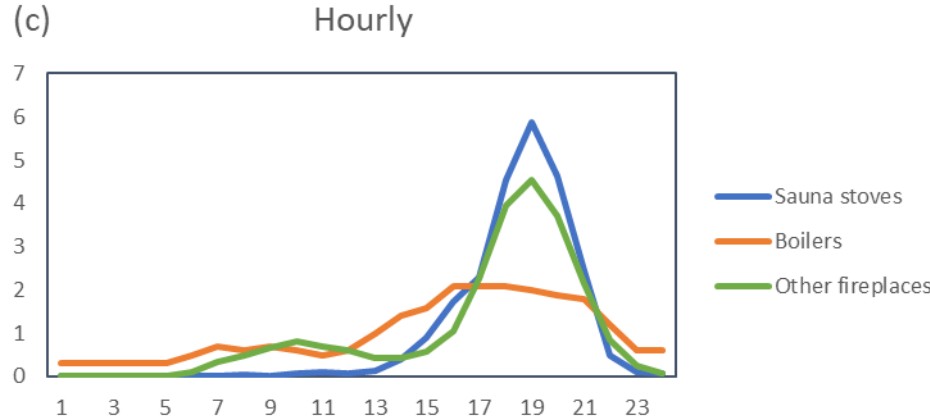

**Figure A1: Temporal monthly, daily and hourly variation coefficients of the RWC emissions for the different categories of wood burning.**



**Appendix B. Evaluation of the representativity of the measurement stations in areas with detached houses**

We have examined the total amounts of emissions originating from RWC in the immediate vicinity (from 50 m to 150 m) of the measurement stations at detached house areas, these are presented in Fig. B1.

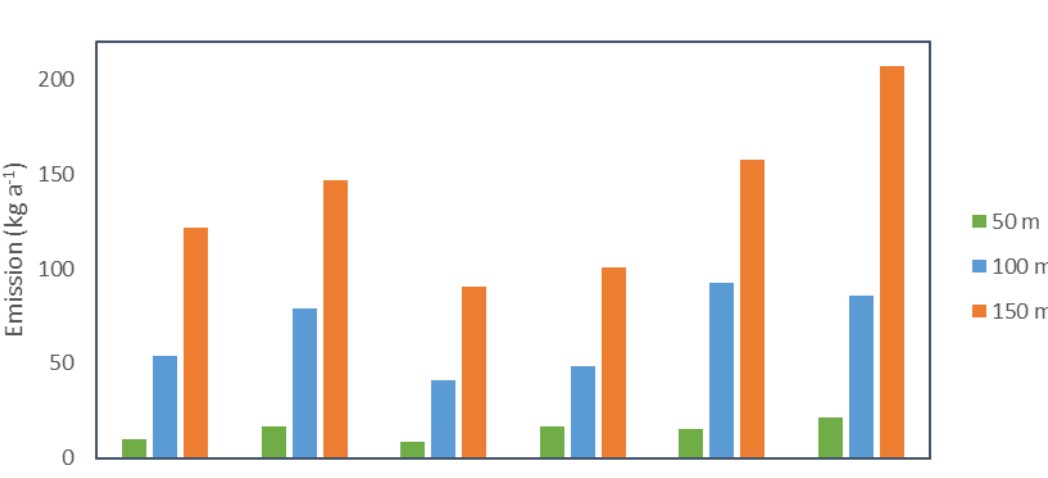

**Figure B1: The total amounts of emissions from RWC near the measurement sites in areas with detached houses, within different**
**distances from the site.**

The stations are exposed to substantially varying amounts of emissions from RWC. For instance, at the site of Ruskeasanta, the amount of emissions within 150 m of the site was more than twofold, compared to the those at stations located in less densely built areas.

**Appendix C. The bias of the model predictions in terms of temperature**

The emissions and dispersion of pollution attributed to RWC depends especially on the ambient temperatures. We have therefore examined the fractional bias values of the predicted and measured concentrations for the winter months, in terms of temperature. These values are presented in Figs. C1a–b and C2a–b. The biases are presented against temperature, for each month, and for the regional and urban background stations and for the station of Vartiokylä, which is the only RWC
measurement station with measurements for several years. For Vartiokylä, we have also compared biases calculated with all winter data with the biases, which were calculated after excluding the local holiday weeks.




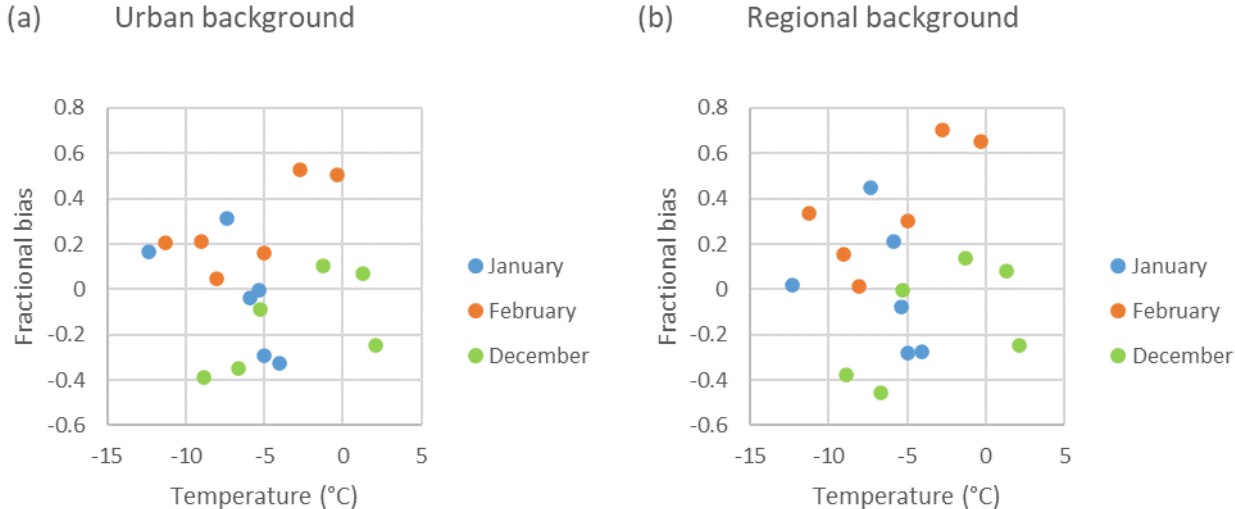

**Figure C1: Fractional bias for the winter months as a function of temperature for background stations. A positive bias corresponds to a model over-prediction.**

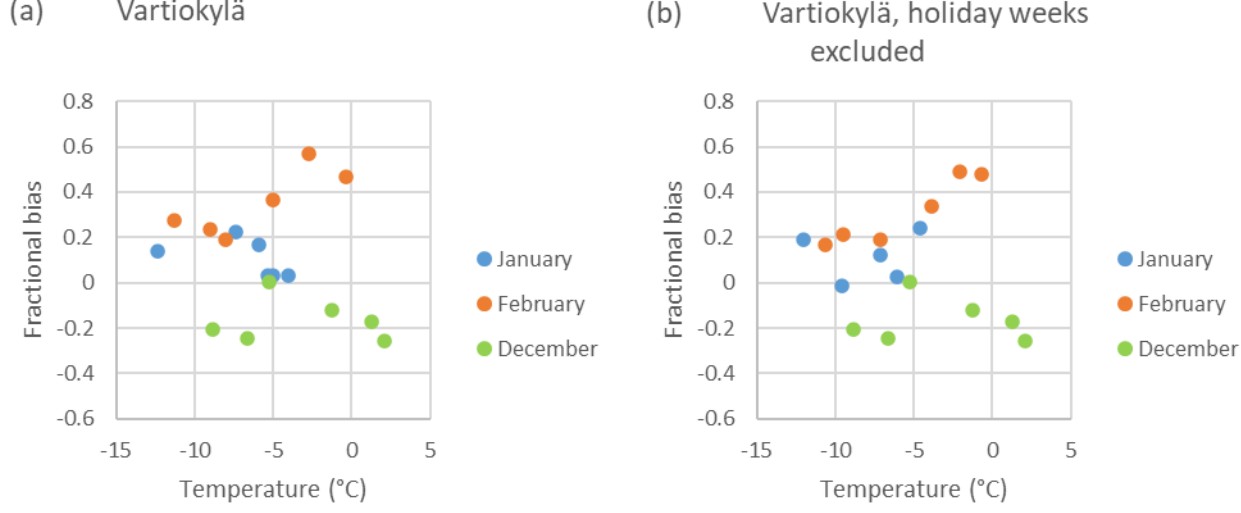

**Figure C2: Fractional bias for the winter months as a function of temperature for the residential station of Vartiokylä. A positive bias corresponds to a model over-prediction.**


For both the background and residential stations, there are slightly more over-predictions in case of the relatively warmer winters. The applied seasonal variation function of the RWC emissions could therefore possibly be adjusted based on these

results, in future research. Especially for February, the dependence of the bias on temperature is more obvious, if we exclude the holiday week from the data. During the holiday week, it is known that a large fraction of people is travelling, and the amount of wood combustion is therefore smaller than during other weeks. We conclude that in addition to the temperature dependence of the emissions from RWC, the impact of other factors, such as local holidays, should be considered.

There have been mostly under-predictions in December at the residential sites. This may have been caused by the traditionally increased recreational wood burning and cooking during the Christmas holiday season. This under-prediction is not correlated with ambient temperature.

**Code and data availability**

The measured and predicted concentration data are available by contacting the corresponding author of this article (leena.kangas@fmi.fi). The SILAM code is available at http://silam.fmi.fi.

**Author contribution**

LK, JK and ArK: planning the research goals, LK: urban scale model computations, evaluation and statistical analysis of the model results, writing the original draft of the article, JK: revision of the text, discussion of the results, MK: contribution to emission and measurement data sets, KR: contribution to post-processing and analysis of concentration data, MS: SILAM model computations, AnK and JVN: compilation of the RWC emission inventory for the Helsinki Metropolitan Area. All authors reviewed and commented on the manuscript.

**Competing interests**

The authors declare that they have no conflict of interest.

**Acknowledgements**

This activity has received funding from the European Union's Horizon 2020 research and innovation program under grant agreement 820655 (EXHAUSTION project). This work reflects only the authors' view, and the Innovation and Networks Executive Agency is not responsible for any use that may be made of the information it contains.

We would also like to thank for the funding of Nordforsk under the Nordic Programme on Health and Welfare (Project #75,007: NordicWelfAir - Understanding the link between Air pollution and Distribution of related Health Impacts and Welfare in the Nordic countries). The funding of the Academy of Finland for the project "Global health risks related to atmospheric composition and weather (GLORIA)" is also acknowledged.



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
