# Peer review of "The contribution of residential wood combustion to the PM2.5 concentrations in the Helsinki Metropolitan Area"

_EGUsphere, 2023_

## Referee Comment (RC1)

**Referee Comment on egusphere-2023-1194 "The contribution of residential wood combustion to the PM2.5 concentrations in the Helsinki Metropolitan Area" by Leena Kangas et al.**

General Comment

The paper presents a multiyear analysis of the contribution of residential wood combustion (RWC) to fine particulate matter (PM2.5) in the metropolitan area of Helsinki. The study builds on a previous study on the contribution of RWC emissions to urban fine particulate matter in Nordic cities (Kukkonen et al., 2020), but extends the scope to the year-to-year and seasonal variations, and a comprehensive validation against measurements.

The study concludes that holiday seasons should be considered when developing the temporal profile of RWC emission, mostly based on a single case when high concentrations were predicted during local holidays. The dynamics of the contribution of RWC emissions should be investigated more systematically given that a large dataset of 6 years is now available. In an upgraded analysis, the authors should compare modelled and measured winter PM2.5 separately for weekdays, for weekends and for holidays. They should also discuss the representativeness of the measurements used for comparison of the model results.

Specific Comments:

1.) Introduction (P2, line 54-58): Please provide increase/decrease of residential biomass consumption as percentage value in the European Union (EU) and for the mentioned European countries.

2.) Introduction (P2, line 58-64): Due to the phase-out of coal in Germany and other countries, there is a trend to replace coal by wood for energy production. Is this additional wood combustion relevant for Helsinki? At least, energy production should be mentioned as potential new urban wood combustion source.

3.) The method to distribute annual RWC emissions over time is not very clear. The basis for the temporal variation seems to be questionnaires for 2013-2014 and a survey on fireplace usage for 2008-2009 (P7, line 168-170). Did the authors use the two surveys separately for different years or did they use the average result of the two studies? It is also not clear whether ambient temperature data was used directly or indirectly for the temporal distribution or not at all. It would be better to use the heating degree-day concept for all kinds of RWC emissions, based on the daily outdoor air temperature, which should be available for any city. The heating degree-day concept has a robust basis, is applied in many air quality models, and is used for the temporal profiles of the CAMS-TEMPO database (Guevara et al., 2021).

4.) RWC emission, after plume rise, were injected at a height of 7.5 m (P7, line 181-184). How was the vertical distribution of RWC emissions evaluated? The exhaust from wood burning should be a warm plume, which rises by buoyancy, in particular in winter when air temperature is below zero. There might be situations of stable inversions in winter as well, but these should not occur so frequently. The injection of RWC emission at one height, without further vertical distribution seems questionable.

5.) Were the regional background concentrations computed with SILAM simply added to the concentrations of the urban dispersion model in the post-processing or were they used as boundary condition for the urban simulation?

6.) Suggest splitting of Table 2 in two parts, the first table for the stations when data is available for all 6 years and the second table for the stations when only one year is available for comparison to observations. The second table could be arranged differently, with the statistical indicators as column headers, and therefore be reduced in space.

7.) P14, line 309-315: How can we be sure that under predicted PM2.5 in summer is not due to missing wood burning sources such as for example sauna stoves?

8.) P17, line 345-352: The predicted high daily concentrations at -4 °C are explained in the text by high predicted RWC emissions during a weekend. The question is then: why did the other weekdays during the winter holiday season not also show high predictions of PM2.5?

9.) Finally, shipping could be a significant contributor to PM2.5 concentrations in the harbor areas of the peninsula as is also stated in the beginning section 2.2. Predicted PM2.5 seems to be rather low in the port areas of the maps in Figure 4. Suggest reminding the reader about the potential ship contribution in section 3.2.

Technical Corrections:

P9, line 252: "Factor-of-two …" I assume this refers to the definition of F2. This should be stated here. F2 is commonly referred to as FAC2. Please add a note if F2 is the same as FAC2.

Figure 3: The coloring in the plots of figure 3 are extremely weak. The colors in the other figures are also quite weak (grey instead of black) but in the scatterplots of this figure, the weak color saturation impedes the reading of the plot and the regression equations.

Page 22, line 431: Replace "clearly higher" by "much higher".

References:

Guevara, M., Jorba, O., Tena, C., Denier van der Gon, H., Kuenen, J., Elguindi, N., Darras, S., Granier, C., and Pérez García-Pando, C.: Copernicus Atmosphere Monitoring Service TEMPOral profiles (CAMS-TEMPO): global and European emission temporal profile maps for atmospheric chemistry modelling, Earth Syst. Sci. Data, 13, 367–404, https://doi.org/10.5194/essd-13-367-2021, 2021.

---

## Author Response (AR1)

**"The contribution of residential wood combustion to the PM2.5 concentrations in the Helsinki Metropolitan Area" by Leena Kangas et al.**

Author's response to referees' comments
28 Nov 2023

In the following text, referee's comments are indicated in green font, author's responses in black font, and descriptions of changes in the manuscript in red font. Line numbers in author's responses refer to the revised marked-up manuscript.

Anonymous referee #1
26 Sep 2023

General Comment

The paper presents a multiyear analysis of the contribution of residential wood combustion (RWC) to fine particulate matter (PM2.5) in the metropolitan area of Helsinki. The study builds on a previous study on the contribution of RWC emissions to urban fine particulate matter in Nordic cities (Kukkonen et al., 2020), but extends the scope to the year-to-year and seasonal variations, and a comprehensive validation against measurements.

The study concludes that holiday seasons should be considered when developing the temporal profile of RWC emission, mostly based on a single case when high concentrations were predicted during local holidays. The dynamics of the contribution of RWC emissions should be investigated more systematically given that a large dataset of 6 years is now available. In an upgraded analysis, the authors should compare modelled and measured winter PM2.5 separately for weekdays, for weekends and for holidays. They should also discuss the representativeness of the measurements used for comparison of the model results.

The original manuscript text in section 3.1.3 indeed gave the impression that the conclusion concerning holidays was based on only one case. Thank you for pointing out this inaccuracy. This is important, as analyzing properly this aspect for RWC has not been presented in previous literature. However, there were actually several analyzed cases pointing in this direction, not just one. We simply chose one particularly clear case as an illustrative example.

The overall conclusion regarding holiday periods in the study is also based on the analysis of biases of monthly averages, which are presented in Appendix C, also with holiday weeks excluded.

We have substantially modified the text in section 3.1.3 (page 20, lines 400-410). We also re-wrote our conclusion in a more moderate way, to correspond better to the nevertheless limited amount of data used.

In the original manuscript, we had not made any comparison separately for weekdays and weekends. Making such an analysis was a good suggestion by the reviewer, and we have now

completed the analysis by adding a further comparison for Vartiokylä, for which a 6-year dataset exists.

We have added evaluation based on biases of average concentrations for weekdays and weekends, as well as holiday weeks and other weeks, in winter. There is a difference between weekends and weekdays in winter, and this comparison also adds evidence to our conclusion concerning holidays. We have presented the evaluation in the revised Appendix C (page 35, lines 651-666), and added text also to Conclusions (page 27, lines 551-553) and to section 3.1.2 (pages 16-17, lines 370-381). We have also specified in more detail which periods we have considered as holidays (Appendix C).

Representativeness of the measurements is indeed an important aspect, and we have discussed it in some detail in sections 2.1.2 (pages 6-7, lines 168-174) and 3.1.1 (page 14, lines 320-322; pages 14-15, lines 332-336), already in the original manuscript. We considered, for instance, the measurement station of Vartiokylä to be better representative for a larger area, compared to the (residential combustion) stations in relatively more densely populated areas. For the latter stations, the station may potentially be located too close to some individual emission source, in view of their representativeness for a wider residential area.

We have also added discussion about these aspects in the conclusions (page 27, lines 528-529).

Specific Comments:

1.) Introduction (P2, line 54-58): Please provide increase/decrease of residential biomass consumption as percentage value in the European Union (EU) and for the mentioned European countries.

The reported changes in the original manuscript referred to contribution of biomass from all fuels in the residential sector.

We have added the percentage values as suggested, using absolute values of emissions taken from Eurostat Data Browser. This part of the Introduction was expanded and also clarified (page 2, lines 54-63).

2.) Introduction (P2, line 58-64): Due to the phase-out of coal in Germany and other countries, there is a trend to replace coal by wood for energy production. Is this additional wood combustion relevant for Helsinki? At least, energy production should be mentioned as potential new urban wood combustion source.

Yes, the Helsinki Metropolitan region also has carbon neutrality goals, and has already applied bioenergy heating plants as one option to replace fossil fuels. It is possible that bioenergy will be further increased in the future.

Discussion on these changes has been added to the text (page 2, lines 71-72).

3.) The method to distribute annual RWC emissions over time is not very clear. The basis for the temporal variation seems to be questionnaires for 2013-2014 and a survey on fireplace usage for 2008-2009 (P7, line 168-170). Did the authors use the two surveys separately for different years or did they use the average result of the two studies? It is also not clear whether ambient temperature data was used directly or indirectly for the temporal distribution or not at all. It would be better to use the heating degree-day concept for all kinds of RWC emissions, based on the daily outdoor air temperature, which should be available for any city. The heating degree-day concept has a robust basis, is applied in many air quality models, and is used for the temporal profiles of the CAMS-TEMPO database (Guevara et al., 2021).

We used the information from both questionnaires, for the whole period. The temporal variation of the emissions of RWC is based on the analysis of the data from both questionnaires. The questionnaire of 2008 – 2009 included more detailed data on monthly, daily and hourly variation of the usage of different types of fireplaces. In the 2013 – 2014 questionnaire, the monthly variation was studied again, to gain supplementary, more up-to-date data. However, the amount of the data did not justify using different temporal variation functions for the different years. The same average temporal variation has therefore been applied for all years.

We have clarified this in section 2.2.1 (page 8, lines 202-203) and in Appendix A (page 28, lines 579-581).

The ambient temperature data was not used in the RWC emission modelling. The heating degree-day would be suitable for boilers, which are used for primary heating purposes, but these are of minor importance in the area. On the other hand, sauna stoves represent a significant part of RWC in the area, and they are heated all year round with no dependence on heating-degree days. Also the use of other fireplaces, which are mostly for recreational purposes, is not clearly dependent on heating-degree days, although it may have some (unknown) dependence on ambient temperature. Therefore, we have decided to use the temporal profiles based on the questionnaires.

We have added explanation about this in section 2.2.1 (page 8, lines 207-212), in Conclusions (page 28, lines 569-572), and in Appendix A (page 30, lines 587-596).

4.) RWC emission, after plume rise, were injected at a height of 7.5 m (P7, line 181-184). How was the vertical distribution of RWC emissions evaluated? The exhaust from wood burning should be a warm plume, which rises by buoyancy, in particular in winter when air temperature is below zero. There might be situations of stable inversions in winter as well, but these should not occur so frequently. The injection of RWC emission at one height, without further vertical distribution seems questionable.

The RWC emissions were treated as area sources in the model computations. The current version of the urban scale UDM-FMI model only allows the use of one average release height for each area source, and this release height is assumed to be independent of weather situation. A vast majority of Gaussian multiple-source urban dispersion models (maybe even all of them) treat area sources this way. It was therefore not technically possible to use a variable release height for each building

or each weather situation, unless we would have made major changes to the dispersion program. In a more detailed computation, in addition to the details of each building, and the plume rise for each of these, one should also take into account the building downwash effect for each RWC source. Downwash can actually be as important as plume rise, clearly, depending on the building and other structures and meteorology. This could be the topic of another article.

For the point sources, the UDM-FMI model includes a plume rise computation.

As the reviewer writes, we have assumed a constant release height of 7.5 m, and this causes uncertainty to model computations. However, we have previously assessed (although not presented in the current manuscript), whether this is a reasonable assumption. We computed the plume rise with the point source model with parameter values of a typical detached house RWC source in this region, and with one-year hourly time series of meteorology representing HMA. The UDM-FMI model also includes the estimation of downwash due to the building and the stack, described in detail in
Karppinen, A., Kukkonen, J., Nordlund, G., Rantakrans, E., and Valkama, I.: A dispersion modelling system for urban air pollution. Finnish Meteorological Institute, Publications on Air Quality 28, Helsinki, 1998.

According to these computations, in the case of a detached house RWC source in this region, it is typical that the outflow velocity of the effluents (sum of the initial velocity and velocity due to buoyancy) is lower (or at least not significantly higher) than wind velocity, and the source height is not significantly above the building height. This ratio of outflow velocity and wind velocity, and the ratio of source height and height of the building, commonly results in downwash due to the building, and effluents are cooled down to ambient temperature and mixed in a cavity area behind the building. In the UDM-FMI model, this downwash case is separately modelled as an area source. In our test run with the point source model, majority of the cases (over 90%) resulted in full downwash, the rest were either partially downwashed or treated as a point source with injection height ranging from 6 to 11 m, with an average of 8 m.

The model also includes further dilution after the initial rise through dispersion parameters. This modelling is described in more detail in Karppinen et al. (1998).

We have revised the presentation of the release height parameter, and included in section 2.2.1 (page 8, lines 221-223) a brief explanation on why this particular numerical value was used.

5.) Were the regional background concentrations computed with SILAM simply added to the concentrations of the urban dispersion model in the post-processing or were they used as boundary condition for the urban simulation?

Background concentrations are added to computed concentrations in the post-processing.

This has been added to section 2.4 (page 10, lines 273-274).

6.) Suggest splitting of Table 2 in two parts, the first table for the stations when data is available for all 6 years and the second table for the stations when only one year is available for comparison to observations. The second table could be arranged differently, with the statistical indicators as column headers, and therefore be reduced in space.

This is a good comment.

We have split the table in two parts.

7.) P14, line 309-315: How can we be sure that under predicted PM2.5 in summer is not due to missing wood burning sources such as for example sauna stoves?

Sauna stoves are a significant emission source throughout the year, and they are included in the emission inventory. However, it is probable that there are other sources which are not included, such as barbeques. This has been mentioned in the text (page 16, lines 359-361).

8.) P17, line 345-352: The predicted high daily concentrations at -4 °C are explained in the text by high predicted RWC emissions during a weekend. The question is then: why did the other weekdays during the winter holiday season not also show high predictions of PM2.5?

We are not totally sure about the reviewer's question. He or she probably means: "… why did the other weekends during the winter holiday season not also show high predictions of PM2.5?" (as the first sentence is about one specific weekend, and then he/she presumably asks about the other weekends, not weekdays, as written).

The model assumes an emission variation, in which the weekend emission is significantly higher compared to weekdays. For the other winter holiday weekends, the model does not show as high deviation from observed concentrations as in 2013. The reason is that in order to have a high predicted concentration, it is also essential that, in addition to high emissions, the weather conditions favour weak mixing of pollutants, i.e., we should have a situation with low wind velocity and high stability, which is not always the case.

We have revised the text in section 3.1.3 (page 20, lines 403-405) in this respect, in response to the reviewer's comment.

9.) Finally, shipping could be a significant contributor to PM2.5 concentrations in the harbor areas of the peninsula as is also stated in the beginning section 2.2. Predicted PM2.5 seems to be rather low in the port areas of the maps in Figure 4. Suggest reminding the reader about the potential ship contribution in section 3.2.

Yes, good point.
We have added some text on this matter in section 3.2 (page 21, lines 445-447).

Technical Corrections:

P9, line 252: "Factor-of-two …" I assume this refers to the definition of F2. This should be stated here. F2 is commonly referred to as FAC2. Please add a note if F2 is the same as FAC2.

Yes, F2 referred to factor-of-two.
We have changed the symbol F2 to FAC2.

Figure 3: The coloring in the plots of figure 3 are extremely weak. The colors in the other figures are also quite weak (grey instead of black) but in the scatterplots of this figure, the weak color saturation impedes the reading of the plot and the regression equations.

We have redrawn figures 2, 3 and 6, and all figures in the appendices.

Page 22, line 431: Replace "clearly higher" by "much higher".

We have corrected this.

References:
Guevara, M., Jorba, O., Tena, C., Denier van der Gon, H., Kuenen, J., Elguindi, N., Darras, S., Granier, C., and Pérez García-Pando, C.: Copernicus Atmosphere Monitoring Service TEMPOral profiles (CAMS-TEMPO): global and European emission temporal profile maps for atmospheric chemistry modelling, Earth Syst. Sci. Data, 13, 367–404, https://doi.org/10.5194/essd-13-367-2021, 2021.

This important reference has been added, and cited in the text in Appendix A (page 30, line 589).

Anonymous referee #2
17 Oct 2023

General comments

The manuscript deals with an important subject; residential wood combustion RWC induced PM pollution in urban areas. Although Helsinki has relatively clean air in general, RWC can make a substantial contribution to PM2.5 concentrations in residential areas during winter months. Given the negative health impacts of long-term exposure to PM2.5 in low concentrations, the general topic is relevant.

The study builds on earlier modelling studies dealing with PM pollution from RWC in Helsinki. The earlier studies dealt mainly with annual level concentrations, while this study concentrated on

validation against monitoring and temporal aspects. Therefore it shows a reasonable level of novelty.

Overall the manuscript is well and clearly written, and the results are presented in a logical manner. It presents concrete conclusions about important future research needs for temporal aspects of emission inventories.

We thank the reviewer for these kind words.

The conclusions chapter itself is longish with repetition about what has been done, and some discussion-like parts with references. Authors might want to consider moving some of these to earlier parts, or adding a Discussions chapter (if the format of the journal allows).

We completely agree on this point. The Conclusions section contained discussions that do not belong there.

We have therefore removed text on earlier research and text on current concentrations from RWC, and rewritten the corresponding texts in the Introduction section (page 4, lines 114-124).

Specific comments

Lines 60-65:

Authors introduce reasons for increase and decrease of RWC in Europe. It is not clear whether the whole chapter is based on Viana et al. or is the vague analysis in the later part the authors' own thinking.

The whole of the chapter was not based on Viana et al.

We have clarified the references in the chapter, and partly rewritten it (page 2, lines 53-63). Also, the speculative estimate on future developments has been removed.

*The growing concern for the health impacts associated with the emissions attributed to RWC has resulted in a consideration of abatement measures for RWC.*
Who has considered, and provide reference or clearly indicate if based on Viana et al.

We agree that this sentence was badly formulated. The abatement has not only been considered, but measures have already been taken, e.g., in the Nordic countries to reduce the emissions of PM2.5, also from RWC.

This has been clarified in the text and some references have been added (page 3, lines 64-66).

*In the long run, this might possibly result in a decreasing trend in RWC, but due to the current requirements to decrease the use of fossil energy in the EU, a significant change is not expected in the near future.*

This gives an impression that the changes in the intensity of use of wood in the residential sector in the EU is mainly policy driven. Deepen the analysis and give support with references.

We have completely rewritten this paragraph and deepened the analysis by describing the factors which possibly have impact on emission changes of RWC, both in the EU and especially in Finland (page 3, lines 64-81).

Lines 153-154:

*It has previously been found that the contribution of other urban source categories to the PM2.5 concentrations has not been significant in this region.*

Provide a reference who has found out. Often other emission sources than the ones mentioned here make a considerable contribution in urban emission inventories. For instance, construction and maintenance activities of housing and streets. To what extent e.g. construction machinery, snow clearing, motorized gardening etc. are included in the vehicular traffic category? And barbeques are mentioned later on.

We have clarified the reference, which concerns power plants and shipping and harbour activities (page 7, lines 182-183).

Construction and maintenance can be a significant emission sources locally, e.g., near major building or road construction sites, but these are of less importance in detached-house residential areas. Unfortunately, there is no reliable inventory or estimate on the effects on air quality of construction and maintenance sites for the HMA.

We have added a discussion of these factors in section 2.2 (page 7, lines 185-189).

Section 3.1.1:
Authors discuss factors that might affect the differences between modelled and measured concentrations. How about the uncertainties related to the spatial distribution of the emissions? How well the actual intensity of the use of RWC inside each 100m grid cell near the monitoring station is known? Although the locations of fireplaces in each house might be known from the house register, the frequency of the use might vary substantially from house to house in supplementary or recreational function. Are there any information about the variability from the surveys and has this been taken into account?

Yes, clearly there are uncertainties related to spatial distribution of emissions. The locations and primary heating methods of individual houses are known. There is no information on the emission intensity of individual houses. However, according to the questionnaire, for instance, the amount of wood used, and the number and type of fireplaces depend on primary heating method. There are also city-specific differences in the amount of wood used. This data has been used in estimating the spatial distribution of emissions. Also, there is no information on the temporal variation of RWC for individual houses, the same average functions have been used throughout the area.

We have added a comment on this uncertainty to the manuscript in section 3.1.1 (page 14, line 325).

*These over-predictions at the residential sites in winter were probably caused by the assumed semi-empirical seasonal variation of RWC emissions in winter; this variation function may not have been ideal for the meteorological conditions during the considered periods.*

What evidence exists that this is the case? Would it be possible to draw out data from questionnaires to support this conclusion?

Evidence for this assumption is based on the more detailed study of the model agreement with observations during the winter months, which is presented in the section 3.1.3 and in Appendix C.

We have clarified the text and partly rewritten the paragraph (page 16, lines 362-370). We have also written the corresponding text in the Conclusions more accurately (page 27, lines 537-541).

*However, the recent deterioration of the economic situation and high energy prices may favour wood burning and increase emissions.*
This is questionable for any mid or longer term prognosis and outdated already for the part of energy prices.

Yes, we agree with the reviewer.

We have therefore removed this sentence. The potential emissions of RWC in the future are discussed in the Introduction (page 3, lines 77-81).

Technical corrections:

Repetition of *clearly*.

This has been corrected.